# Microencapsulates of Blue Maize Polyphenolics as a Promising Ingredient in the Food and Pharmaceutical Industry: Characterization, Antioxidant Properties, and In Vitro-Simulated Digestion

**DOI:** 10.3390/foods12091870

**Published:** 2023-04-30

**Authors:** Nada Ćujić Nikolić, Slađana Žilić, Marijana Simić, Valentina Nikolić, Jelena Živković, Smilja Marković, Katarina Šavikin

**Affiliations:** 1Department for Pharmaceutical Research and Development, Institute of Medicinal Plants Research “Dr Josif Pančić”, Tadeuša Koščuška 1, 11000 Belgrade, Serbia; ncujic@mocbilja.rs (N.Ć.N.);; 2Laboratory of Food Technology and Biochemistry, Maize Research Institute, Zemun Polje, Slobodana Bajića 1, 11185 Belgrad-Zemun, Serbia; 3Institute of Technical Sciences of SASA, Knez Mihailova 35/IV, 11000 Belgrade, Serbia

**Keywords:** blue maize, waste, anthocyanins, spray drying, biopolymers, microparticle size, HPLC

## Abstract

An anthocyanin-rich blue maize waste product was used for anthocyanin extraction. To preserve bioactive phenolic compounds, a spray-drying technique was employed using conventional wall material maltodextrin (MD), with novel one, hydroxypropyl-β-cyclodextrin (HPBCD). The obtained spray-dried maize extract (SME) and microencapsulates were analyzed based on physicochemical powder properties, chemical analysis, antioxidant activity, and digestibility. The examined microencapsulates demonstrated good powder properties, exhibited a high powder yield (up to 83%), and had a low moisture content (less than 5%). HPBCD and MD + HPBCD combinations demonstrated superior powder properties in the terms of decreasing the time necessary for rehydration (133.25 and 153.8 s, respectively). The mean average particle diameter ranged from 4.72 to 21.33 µm. DSC analyses signified high powder thermal stability, around 200 °C, related to the increasing preservation with biopolymer addition. The total phenolic and anthocyanin compounds ranged from 30,622 to 32,211 mg CE/kg (CE—catechin equivalents) and from 9642 to 12,182 mg CGE/kg (CGE—cyanidin 3-glucoside equivalents), respectively, associated with good bioactive compound protection. Microencapsulates with both carriers (15% MD and 15% HPBCD) had the highest digestibility (73.63%). Our results indicated that the microencapsulates created with the active ingredient and the wall materials (MD and HPBCD) could protect phenolic compounds/anthocyanins against ABTS radicals (63.53 and 62.47 mmol Trolox Eq/kg, respectively).

## 1. Introduction

Anthocyanins (ACNs) are soluble free phenolic compounds responsible for the blue, red, and purple colors primarily in fruits and vegetables. In addition to their important physiological functions in plants, ACNs may be used in the food and pharmaceutical industry, having both coloring and antioxidant impacts. In general, ACNs are widely used by humans, with an estimated daily intake about of 12.5 mg/day in the United States [1]. Further, many studies indicate that the utilization of foods high in ACNs has been linked to lower risks of chronic disorders such as obesity and diabetes [2,3], cardiovascular diseases [4,5], cancer [6,7], and eye diseases [8]. Wu et al. [1] screened more than 100 commonly used foods (fruits—29, vegetables—26, nuts and dried fruits—14, spices—15, other food—19) for ACNs. Twenty-four of them were found to contain ACNs. The content of total anthocyanins varied from 7 to 14,800 mg/kg of fresh weight in gooseberry and chokeberry, respectively. The composition of ACNs in foods is quite different as well [1]. Cereal grains can also be a source of ACNs. Several commercially applicable cereal genera such as maize, wheat, rice, oat, barley, sorghum, and millet contain some noticeably colored varieties [9,10]. ACNs are localized in the outer layers of cereal grains. In purple and blue maize genotypes, ACNs are found in the pericarp and aleurone layer, respectively, while in black, dark red, and deep purple maize grains, they are found in both layers. The total ACN content in maize grains, their composition, and their possible application have been published in numerous studies [10,11,12,13,14]. According to the results of Žilić et al. [11,14], the content of total anthocyanins varied from 2.5 mg CGE/kg in red–yellow maize grain to 4888.9 mg CGE/kg in grains of a deep purple maize genotype. Ten and seven ACNs have been found in blue and deep purple maize, respectively. Cyanidin derivatives were found as dominant, and their acylated forms accounted for about 98 and 29% of the total ACN content in the grains [14]. Whole-grain flour of colored maize has been primarily used in bakery and confectionery production. However, the potential application of ACNs originating from maize grains in dietary supplements, as food additives and related products, requires their isolation and represents a challenge attributable to their limited stability against factors such as light, oxygen, low or high temperature, and different pH values. One approach to overcome this difficulty is the encapsulation process of bioactive compounds, especially in micro-sized forms. In addition to the whole grain, the relatively large amount of waste generated during the processing of colored maize can also be a rich source of anthocyanins. Microencapsulation can be considered a suitable technology that could increase polyphenol stability throughout the storage time, obtaining a product with a longer shelf-life as well as increased bioavailability [15,16]. Among many examined microencapsulation procedures, spray drying can be promoted as a cost-effective, adaptable, appropriate, and widely used method for the production of good-performing micro-sized powders in foods and pharmaceuticals [17,18]. The stability and oral bioavailability of phenolic compounds could be improved at the same time by using different carriers to enhance their biological activities [19,20]. Suitable carriers for the microencapsulation process are natural biopolymers, which are commonly used, and well-known polysaccharides such as maltodextrin (MD) [19,21,22,23,24,25]. Besides the conventional wall material, such as maltodextrin, innovative materials, such as cyclodextrins (CDs), have been proposed to overcome the limitations of polyphenols [23,26]. Cyclodextrins are natural cyclic compounds that can encapsulate a wide range of bioactive compounds [26,27,28]. Among them, hydroxy-propyl-β-cyclodextrin (HPBCD) has commercial availability, a low cost, and a suitable cavity size, which is suitable for different active compounds. HPBCD is the first approved cyclodextrin derivative according to Food and Drug Administration and belongs to the GRAS (generally recognized as safe) list. The synergy of commonly used biopolymers with novel ones in double microencapsulates, which could entrap the bioactives, could be a promising perspective from the standpoint of food and pharmaceutical applications [23,26,27,28].

The research goal was to develop microencapsulation produces for phenolic compounds from blue maize processing by-products using conventional wall material, MD, in combination with a novel material, HPBCD. Microencapsulates were characterized to obtain systems with appropriate functional, organoleptic, and biological characteristics, as well as a uniform size, which could be candidates in food and pharmaceutical applications. There are many reports of spray-drying applications in food industries, especially in the production of microparticles extracted from fruit such as chokeberry [29,30], blueberry [31], and blackberry [32]. According to our knowledge and the literature, the microencapsulation of anthocyanin-rich extracts obtained from blue maize by-products after processing was conducted for the first time.

## 2. Materials and Methods

### 2.1. Materials

#### 2.1.1. Blue Popping Maize By-Product 

The by-product in the process of grinding blue popping maize grains was used in this study. The genotype of blue popping maize is an old maize variety collected in the vicinity of Kragujevac (central part of Serbia). It is kept in the gene bank of the Maize Research Institute Zemun Polje (MRIZP), Serbia. Previous research has shown that grains are rich in ACNs. The total ACN content was 907.51 mg CGE/kg with (cyanidin 3-(6′-malonyl glucoside) (Cy-3-6Mal-Glu) being dominant [14] (Appendix A). During the grinding of blue popping maize grains in the production plant of the MRIZP, flour with fine particles (212 µm) accumulates as waste, in the amount of about 20% of the total, on the protective sieve and parts of the processing equipment. The content of total ACNs in this flour is on average 1360 mg CGE/kg. Considering that it is a waste rich in bioactive compounds, the fine powder was used for the extraction of free phenolic compounds and the preparation of microencapsulates.

#### 2.1.2. Biopolymers (Carriers)

Maltodextrin (MD) (DE 17-20) was obtained from Davisco Foods International (Le Sueur, MN, USA), and hydroxypropyl-β-cyclodextrin (HPBCD) (97% grade) was obtained from Acros Organics (Geel, Belgium).

### 2.2. Preparation of Blue Maize By-Product Extract

The waste by-product of blue maize processing was used for the extraction process. The optimization of the extraction process was performed as preliminary screening to choose the optimal extract. The extraction process consisted of the maceration method on an orbital shaker (Unimax 1010, Heidolph, Schwabach, Germany, 180 rpm, 25 °C, for 1 h). To determine the influence of total polyphenols and ACN extraction, four levels of solvent (ethanol–water) (0, 20, 30, and 50%) and two levels of the solid–solvent ratio (1:5 and 1:10) were used. Ethanol (50% *w*/*w*) and a solid–solvent ratio of 1:10 were chosen as optimal and were used in the percolation process for the production of a higher amount of extract. Subsequently, ethanol was vaporized by a rotary evaporator (Buchi rotavapor R-114) under vacuum at 50 °C to preserve the sensitive phenolic compounds, especially ACNs and dealcoholized liquid blue maize extract (LME), which were further subjected to microencapsulation.

### 2.3. Microencapsulation by the Spray-Drying Method

Liquid blue maize extract (LME) was spray-dried with and without carrier agent addition. Microencapsulates were prepared using MD (30%, *w*/*w*), HPBCD (30%, *w*/*w*), and a combination of both carriers (15% MD and 15% HPBCD). The addition of carriers used in the experiments was established based on the LME dry weight. Biopolymers were independently dissolved in previously prepared LME prior to the microencapsulation process while HPBCD was dissolved 24 h before the spray-drying process to enable micellization [33]. For a combination of the two carriers, MD was added to the HPBCD solution after 24 h. Before the spray-drying process, the biopolymer–extract mixtures were prepared at 40 °C under constant agitation by a magnetic stirrer. Completely homogenized solutions were treated with a Labtex ESDTi spray dryer (Labtex, Huddersfield, UK), with a 0.5 mm nozzle diameter, under the following conditions: inlet (135 ± 10 °C) and outlet (65 ± 10 °C) temperatures, air flow rate (70 m^3^/h), liquid feed (11 mL/min rate), and atomization pressure (2.5 bar). The four obtained powders, spray-dried maize extract (SME), and microencapsulates, were kept in high-density glass bottles in a desiccator at room temperature prior to analysis.

### 2.4. Analysis of the Technological Characteristics of SMEs and Microencapsulates

#### 2.4.1. Powder Yield

The powder yield (Y) was calculated as the ratio between the obtained SME mass (g) at the end of the spray-drying process and the expected mass:Y (%) = m_extract_/m_expected_ × 100

The estimated (expected) mass was the mass of LME dry residue multiplied by the LME mass exploited for the drying process and the mass of the used biopolymer:m_expected_ (g) = m_biopolymer_ + m_dry residue_ × mLME

#### 2.4.2. Moisture Content

The water contents (moisture vs. dryness) of each powder were examined. Powders were heated at 105 °C until constant weight by a Halogen Moisture Analyzer HB43-S (Mettler Toledo, Columbus, OH, USA). Achieved results are reported as a percentage (%).

#### 2.4.3. Bulk Density

Bulk density determination was achieved based on the method previously reported by Vidović et al. [34] with minor alterations [35]. A measured mass of 1 g of a piece of sample was placed in a 5 mL graduated glass cylinder and shaken at room temperature for 5 min with agitation fixed at 300 rpm. Subsequently, volumes of dried samples in a glass cylinder were measured directly from the cylinder. Bulk density was determined as the ratio of the powder mass and determined volume of dry powder, represented as milligrams of dry powder per milliliter (mg/mL). The cylinders were tapped 120 times, and the powder volume was determined for the powder-tapped densities. The values of flowability and cohesiveness of the examined samples were determined as the Carr index (CI) and Hausner ratio (HR), calculated using the following equations:CI = (ƍ_tapped_ − ƍ_bulk_)/ƍ_tapped_ × 100
HR = ƍ_tapped_/ƍ_bulk_

#### 2.4.4. Rehydration

The spray-dried sample rehydration time can be considered to be a period necessary for the complete dissolution of the powders in water at room temperature. It has been determined as the time for the full reconstitution of 1 g of dried sample in 50 mL of water, using a magnetic stirrer, expressed in seconds (s) [36]. For each dissolved sample, the pH value was determined using a pH meter (Hanna HI 99161, Portugal).

### 2.5. Physical Characterization of SME and Microencapsulates

#### 2.5.1. Particle Size Distribution

The particle size distribution of the examined samples was determined using a Mastersizer 2000 analyzer (Malvern Instruments, Worcestershire, UK) [29,37]. Parameters d_10_, d_50_, and d_90_, which represent 10%, 50%, and 90% of the particles that were smaller than the remaining ones, were verified. The indicator of the particle size distribution width was defined as the SPAN value (or PSD), estimated as (d_90_ − d_10_)/d_50_. D [3.2], the surface-weighted mean, and D [4.3], the volume-weighted mean, as well as the uniformity of the microparticles were examined. 

#### 2.5.2. FTIR Spectroscopy

The Fourier-transform infrared (FTIR) spectra of the spray-dried powders (pure dried extract: SME and microencapsulates: SME + biopolymers) were obtained between 400 and 4000 cm^−1^, with a resolution of 4 cm^−1^ by a Nicolet iS10 FTIR spectrometer (Thermo Scientific, Gothenburg, Sweden) [37]. The FTIR data were recorded on directly applied samples, and the spectral range was measured for three replicates (each sample was analyzed separately).

#### 2.5.3. Differential Scanning Calorimetry (DSC) Analysis

The thermal properties of the spray-dried powders were evaluated by a DSC131 Evo (SETARAM Instrumentation, Caluire-et-Cuire, France) according to a previously published method [38]. Powders were placed in aluminum pans (30 µL) and hermetically packed. A blind probe was the empty pan. The heating process was the following: the reference and sample pans were stabilized at 20 °C for 5 min, after which the temperature was increased to 200 °C at a 10 °C/min heating rate, and the nitrogen flow was 20 mL/min. A baseline run was performed using empty pans under the same conditions, whereas baseline subtraction and enthalpy (J/g) were determined using CALISTO PROCESSING software provided by SETARAM Instrumentation.

### 2.6. Chemical Analyses of SME and Microencapsulates

#### 2.6.1. Extraction of Soluble Free Phenolic Compounds 

For the detection of total phenolic compounds and phenolic acids, extracts were prepared by continuous shaking of 0.3 g of sample (SME and microencapsulated SMEs) in 5 mL of 70% (*v*/*v*) acetone for 30 min at room temperature. The supernatant was used for experiments after centrifugation at 8000 rpm for 3 min at 4 °C. For the detection of soluble free phenolic acids, extracts (5 mL) were evaporated at 30 °C under a N_2_ stream to dryness, and the final residues were redissolved in 1.5 mL of methanol [11]. The prepared methanolic solutions were used for the HPLC analysis. The extracts were kept at −70 °C prior to analysis. All extractions were performed in duplicate for each sample.

#### 2.6.2. Analysis of Total Phenolic Compounds

The total phenolic content was determined by the Folin–Ciocalteu assay [39] and expressed as mg of catechin equivalent (CE) per kg of dry matter (d.m.). Briefly, 10 μL of extracts was adjusted to 500 µL with distilled water after 250 μL 0.2 M Folin–Ciocalteu reagent was added. The mixture was neutralized with 1.25 mL of 20% aqueous Na_2_CO_3_ solution after 5 min, and the absorbance was measured at 725 nm after 40 min.

#### 2.6.3. Analysis of Phenolic Acids by HPLC

The extracts were filtered through a nylon syringe filter of 0.45 µm. Phenolic acids were separated on a Hypersil GOLD aQ C18 column (150 mm × 4.6 mm, i.d., 3 μm) using a linear gradient elution program. The mobile phase contained solvent A (1% formic acid) and solvent B (100% methanol). The flow rate was 0.8 mL/min at 25 °C. The solvent gradient was programmed as described by Žilić et al. [11]. Chromatographic analyses were performed on a Thermo Scientific Ultimate 3000 HPLC with a photodiode array detector. The chromatograms were recorded at 280 nm. Standards of gallic acid, 3,4-dihydroxybenzoic acid, chlorogenic acid, vanillic acid, caffeic acid, syringic acid, *p*-coumaric acid, sinapic acid, ferulic acid, and isoferulic acid were used (10, 20, 40, 50, and 100 µg/g). Identified phenolic acid peaks were confirmed and quantified using Thermo Scientific Dionex Chromeleon 7.2. chromatographic software. The results are expressed as µg per g of d.m.

#### 2.6.4. Analysis of Total Anthocyanins

Total ACNs were extracted from 100 mg of SME and microencapsulated SMEs by mixing with 10 mL of methanol acidified with 1 M HCl (85:15, *v*/*v*). After shaking for 30 min at ambient temperature, the crude extract was centrifuged at 8000 rpm for 3 min at 4 °C. Absorbance was measured at 535 and 700 nm to detect anthocyanins. All extractions were performed in duplicate for each sample. The content of total anthocyanins was calculated using the molar extinction coefficient of 25,965 Abs/M × cm and a molecular weight of 449.2 g/mol and expressed as mg of cyanidin 3-glucoside equivalent (CGE) per kg of d.m. [40]. The prepared extracts were also used for the analysis of individual anthocyanins by HPLC.

#### 2.6.5. Analysis of Individual Anthocyanins by HPLC and HPLC-MS

Individual anthocyanins were determined by HPLC and HPLC-MS analyses under the conditions described in the paper by Žilić et al. [14]. Pure anthocyanin compounds such as De-3-Glu (delphinidin-3-glucoside), Cy-3,5-diGlu (cyanidin-3,5-diglucoside), Cy-3-Glu (cyanidin-3-glucoside), Pt-3-Glu (petunidin-3-glucoside), Pg-3-Glu (pelargonidin-3-glucoside), Pn-3-Glu (peonidin-3-glucoside), and Mv-3-Glu (malvidin-3-glucoside) were used as references for concentration, retention time, and characteristic UV. Non-acylated anthocyanin peaks identified by HPLC analysis were confirmed and quantified using Thermo Scientific Dionex Chromeleon 7.2. chromatographic software, and the results are expressed as μg per g of d.m. The remaining compounds (acylated anthocyanins) were tentatively identified using a combination of the retention time, peak spectra, mass-to-charge ratio, and pattern of fragmentation. Samples were injected into the Waters HPLC system consisting of 1525 binary pumps, a thermostat, and a 717+ autosampler connected to the Waters 2996 diode array and EMD 1000 single quadrupole detector with ESI probe (Waters, Milford, MA, USA). After comparison of the data obtained from HPLC and HPLC-MS analyses, acylated derivatives were quantified using HPLC peak area values (mAU*min) and external standard curves for Cy-3-Glu. Stock standard solutions were prepared in methanol acidified with 1 M HCl (85:15, *v*/*v*) at a concentration of 1.0 mg/mL. The working solutions were prepared by diluting the stock solutions with acidified methanol to concentrations of 5.0, 10.0, 20.0, 40.0, 50.0, and 100.0 μg/mL. The content of acylated derivatives of cyanidin was calculated as equivalent to their glucoside form and expressed as μg per g of d.m.

#### 2.6.6. Analysis of the Total Antioxidant Capacity

According to the direct or QUENCHER method described by Serpen et al. [41], the antioxidant capacity of SME and microencapsulated SMEs was measured. ABTS (2,2-azino-bis/3-ethil-benothiazoline-6-sulfonic acid) was used. The total antioxidant capacity was expressed as the Trolox equivalent antioxidant capacity and presented as mmol of Trolox per kg of d.m. (mmol Trolox Eq/kg).

#### 2.6.7. In Vitro Multistep Enzymatic Digestion Protocol

To determine the digestibility potential of SME and encapsulated SMEs, an in vitro multistep digestion procedure was applied. The method, consisting of oral, gastric, duodenal, and colon phases, proposed by Papillo et al. [42] and modified according to Hamzalioğlu and Gökmen [43], was performed without an attempt to quite mimic gastrointestinal digestion. As a way of mimicking the conditions present in the human gastrointestinal tract, digestion fluids simulating the saliva (simulated salivary fluid, SSF), gastric juice (simulated gastric fluid, SGF), and duodenal juice (simulated duodenal fluid, SDF) were used. These digestion fluids were prepared according to Hamzalioğlu and Gökmen [43]. Samples (5 g) were measured for analysis. SSF solution was used to simulate the oral passage. A mixture of SGF and pepsin solutions (12.5 mg/mL pepsin in 0.1 M HCl) (Pepsin from porcine gastric mucosa, specific activity ≥250 units/mg solid), adjusted to pH 2.0 and incubated at 37 °C for 2 h, was used to simulate the gastric phase. For simulation of the duodenal phase, a mixture of SDF solution with bile salts (10 mg of bile salts per mL of SDF) and pancreatin solution (10 mg/mL pancreatin in distilled water) (pancreatin from porcine pancreas, specific activity 4 × USP specifications) adjusted to 7.5 were used. The simulated duodenal phase was also achieved by incubation of the sample and solution mixture at 37 °C with constant shaking for 2 h. The colon phase was simulated when protease solution (1 mg protease per ml of water) (Protease from *Streptomyces griseus*, specific activity ≥3.5 units/mg solid) was added, and the mixture was incubated at 37 °C by shaking for 1 h, after the pH was adjusted to 8.0. Lastly, Viscozyme L with specified enzyme activity ≥100 FBGU/g was added, and the mixture was incubated at 37 °C by shaking for 16 h, after adjusting the pH to 4.0. The activities of the applied enzymes were not measured after completion of the digestion protocol given that their respective activities were pH-dependent and directed towards the degradation of specific groups of compounds in subsequent steps of the digestion protocol. After multistep enzymatic digestion, the samples were filtered through qualitative paper, air dried in a ventilation oven for 2 h, and subsequently dried at 105 °C for 4 h to constant mass. The dry residue was measured. The digestibility was calculated as a percentage of the difference between the dry sample before the applied protocol and the dry residue.

### 2.7. Statistical Analysis

The experimental data are presented as the mean ± standard deviation of at least two independent repeats. Obtained results were statistically analyzed by Statistica software version 5.0 (StatSoft Co., Tulsa, OK, USA). Significances of differences among samples were analyzed by Tukey’s test. Differences at *p* < 0.05 are considered significant.

## 3. Results and Discussion

### 3.1. Technological Properties of SME and Microencapsulates

#### 3.1.1. Powder Yield

The powder yield of the drying vs. microencapsulation process is an important parameter associated with the process efficiency. The spray-dried maize extract had a yield of 73% (Table 1). Although the most effective carrier was MD, which produced a yield of around 83%, there was no significant difference regarding carrier addition. Similar observations with MD were reported by a group of authors [34,44,45,46]. The usage of HPBCD had a similar yield as pure SME, while the biopolymer combination (MD + HPBCD) slightly reduced the powder yield. In general, the yield, as the effectiveness of the drying process based on established laboratory dryers, ranges from 20 to 70% due to the powder losses and adherence to the drying chamber and cyclone walls [35,47]. All obtained samples exhibited a high powder yield (higher than 50%), which can be considered a reference value for an efficacious drying process, appropriate for lab-scale applications [48].

#### 3.1.2. Moisture Content

Drying effectiveness or moisture content as an important parameter of powder quality demonstrates many effects on the technological characteristics, shelf life, and packaging of powders and represents a crucial impact on a powder’s stability. This powder parameter can be influenced by different factors (carrier type and concentration, temperature condition during the drying process) [49]. Selected drying conditions enabled the preparation of powders with a moisture content of less than 5% (for pure SME, slightly higher than 5%), considering them dry products (Table 1). Biopolymer addition decreased the moisture content. Similar results were confirmed by other groups of authors [34,50]. The moisture content of the powder obtained with HPBCD addition had the lowest value according to the results reported by Wilkowska et al. [31]. Pasrija et al. [51] demonstrated the opposite results with obtained green tea powders. Namely, for all trough spray-drying processes of green tea extract, the highest moisture content was achieved using a similar biopolymer, beta-cyclodextrin, while the maltodextrin/cyclodextrin synergistic effect was caused by the intermediate moisture content, similar to our study. Escobar-Avello et al. [33] obtained a powder with a higher moisture content using MD + HPBCD combinations compared to unencapsulated grape cane extract. The lowest moisture content in powder was obtained using 30% MD by Pudziuvelyte et al. [52], indicating a similar pattern as that in our study. The low moisture content and high production yield of the obtained samples are associated with an extended shelf life and quite stable powders in microbiological terms.

#### 3.1.3. Bulk and Tapped Densities, Flow, and Cohesiveness of Powder Properties

The quality parameters of powders that are predictors of their application such as the bulk and tapped densities, Carr index (CI), and Hausner ratio (HR) were determined and the results are presented in Table 1. The powder bulk and tapped densities are important criteria that can determine the quality of the final products. Bulk density values ranged from 0.260 to 0.374 g/ mL, and powder without carrier addition had the lowest bulk density. The addition of MD increased this property compared to the unencapsulated one, and a similar pattern was reported by Sahin-Nadeem et al. [45]. Tapped density ranged from 0.377 for SME + MD to 0.454 for SME + HPBCD samples. Relatively low values for bulk and tapped densities could be attributed to regular and uniform particles produced via spray drying as a microencapsulation technique, which was confirmed by the laser diffraction method. 

The particle flow (CI) and cohesiveness characteristic (HR) of spray-dried microencapsulates were evaluated and classified according to a previously reported study by Caliskan and Dirim [35]. The CI of spray-dried blue maize extracts ranged from 9.17 to 29.29, justifying the findings that SME + MD, as well as SME + MD + HPBCD, could be considered powders with very good properties (less than 15), followed by SME + HPBCD, with 17.51, as a powder with good properties. All powders microencapsulated with carrier addition could be viewed as powders with good flowability, without statistical significance. The powder cohesiveness based on the Hausner ratio corresponded to low (value of 1.2 and less), based on literature data by Caliskan and Dirim [35], for SME + MD, SME + MD + HPBCD, followed by SME + HPBCD. Biopolymer involvement caused better flowability and less cohesive behavior of the powders, MD and MD + HPBCD demonstrated superior powder properties. However, pure SME had higher cohesiveness and a lower ability to flow freely. 

#### 3.1.4. Rehydration and pH

Time for powder reconstitution vs. rehydration is the time needed for the entire dissolution of a specified quantity of powder in a suitable medium. The time required for powder reconstitution has practicable significance for instant dried preparations or reconstructed beverages. The rehydration time varied between 133.25 (SME + HPBCD) and 224.2 s (pure SME) (Table 1). However, pure SME had the highest (longest) rehydration time according to the highest flowability. Biopolymer addition had an influence on the rehydration time. HPBCD and MD + HPBCD combinations demonstrated superior powder properties in terms of decreasing the time necessary for rehydration. Since MD is starch and HPBCD is a more soluble derivative, the results for the rehydration time in this study were expected and are similar to those obtained by other authors [31,53,54]. The pH values of the reconstituted powders were around 6.5, without statistical significance, which is appropriate for oral administration. 

### 3.2. Physical Characterization of SME and Microencapsulates

#### 3.2.1. Particle Size Distribution

Figure 1 and Table 2 show the particle size distribution of spray-dried samples produced using different biopolymers. The modal distribution of the particle size for all obtained spray-dried formulations was demonstrated (Figure 1), with three distinct peaks representing predominant sizes. The diameter of spray-dried microparticles varied from 2.22 (d_10_) for SME + MD to 257.14 µm (d_90_) for SME + MD + HPBCD, with the mean average diameter d_50_ from 4.72 to 21.33 µm for all microencapsulated powders (Table 2). Pure SME exhibited the highest particle diameter, likely related to the highest moisture content. The microparticles produced with MD had significantly lower mean diameters. Generally, carrier addition decreased the particle size compared to pure SME. Parameter D [4.3] (mean diameter over the volume distribution) was in the range of 32.34 for SME + MD to 78.53 µm for the pure spray-dried extract. Similar results of the mean diameter, determined using Scanning electron microscopy, were reported by Wilkowska et al. [31], using 15% MD and 15% HPBCD for the spray drying of blueberry juice, with a slightly higher diameter dispersing nozzle (0.7 mm), even if similar inlet and outlet temperatures were applied. Pasrija et al. [51] demonstrated the increase in particle size with CD addition compared to MD during the microencapsulation of green tea extract, whereas the MD + CD combination had a lower particle diameter. Escobar-Avello et al. [33] found a reduction in mean particle size with the addition of MD and HPBCD in grape cane extract. Particle size distribution (PDI) was rather broad, characterized by high span values, and the most homogenous was pure SME, followed by SME + HPBCD. The differences in PDI between samples might be related to polymer properties [55]. The PDI possessed higher values as inclusion complexes with MD and HPBCD, probably due to the HPBCD agglomeration tendency, a consequence of cyclodextrin self-assembly in water [56]. The particle size values of spray-dried formulations in this study can be considered well-formed microencapsulates [19,57]. The mean diameter of the obtained powders indicates the formation of small and uniform particles appropriate for food and pharmaceutical applications. Particles with smaller diameters could positively affect the solubility of the powders and the sensory characteristics of the products, while larger ones can cause undesirable agglomeration processes.

#### 3.2.2. FTIR Spectroscopy

FTIR analysis is a convenient technique to detect the inclusion of complex formations and relevant changes in spectra (reduction, disintegration, absorption bands, and intermolecular interactions). This method can provide some additional information about microencapsulated guest molecules and polyphenol compounds in selected biopolymers. The infrared spectra of the spray-dried blue maize extract and microencapsulates are shown in Figure 2. All of the observed spray-dried microencapsulates had similar spectra. The spectrum of the blue maize extract exhibited the highest absorption band, between 3600 and 3000 cm^−1^, originating from O–H (hydroxyl) groups associated with phenolic compounds, while the band at 2900 cm^−1^ has been correlated with C-H stretching [33]. The peaks at 1600 cm^−1^ originated from C=C stretching bands, while COO- vibrations are due to polysaccharide molecules [58]. The spectra contained typically strong absorption bands of carbohydrates in the region of 1000 cm^−1^, related to C-H and C-O aromatic ring stretching, signifying the aromatic compound existence originating from phenolics in the extract [29,44]. The FTIR analysis also corresponds to the analysis of possible interactions among the polymers and functional ingredients. In general, it was observed that blue maize extract microencapsulation along with suitable biopolymers such as MD and HPBCD did not create meaningful changes, which is, according to the literature, an indication of a successful microencapsulation process [59]. This property denotes that chemical bonds specific to functional ingredients such as blue maize extract and wall materials were preserved, and microencapsulation resulted from physical interactions rather than chemical ones. Therefore, the spray-drying process did not transform the polymer matrix and extract structures, and the FTIR method indicated that microencapsulation developed from physical incorporation. Insignificant alterations in intensity were noted, implying the microencapsulation of phenolic compounds in two different biopolymers without interactions between them.

#### 3.2.3. Thermal Characteristics based on DSC Analysis

With the goal of the thermal stability examination of the blue maize microencapsulates, the DSC analysis was performed. DSC thermograms were recorded for pure carriers, pure SME, as well as extract and carrier combinations. Along with FTIR analysis, the DSC method can monitor the interactions between encapsulated phenolic compounds and carriers. The glass transition temperature of the examined samples could be caused by different factors, chemical compositions, molecular weights, and moisture contents [60]. The peak transition temperatures of the SME and microencapsulates, as well as enthalpy changes (∆H), are demonstrated in Table 3. The thermogram showed the first stage of analysis, where the peak maximum of all microencapsulated samples was around 80 °C, without statistical significance, indicating a similar behavior of powders (Table 3). The DSC curves in this phase were characterized by an initial broad endotherm peak, probably associated with the evaporation of absorbed water, structural water, and volatile compounds [19,50]. The low energy of this zone is due to the low moisture content of the powders. The desorption of structural water in this phase and the lowest peak temperature and enthalpy change were noticeable with SME + HPBCD, corresponding to the lowest moisture content of this powder. During the second stage, changes were observed around 150 °C, and samples underwent a greater enthalpy change, probably due to polyphenol degradation and the melting of polymers, indicating again the highest value of energy necessary to provoke the change in SME + HPBCD, but without statistical significance. The experimentally determined transition temperature of the pure HPBCD was in the range between 28 and 153 °C, and for spray-dried SME + HPBCD, 40 to 220 °C, during the first two stages of transition. Carbohydrates, such as MD and HPBCD, preserved the polyphenolic structure, as reported by Jovanović et al. [47]. In the third phase, the temperature ranged from 197 °C for SME to around 240 °C for microencapsulated powders, which could be associated with self-degradation. 

DSC diagrams (Figure 3) almost exclusively demonstrated extensive and broad endothermic peaks between 40 and 200 °C, which can be attributed to the sum of glass transition and melting. Comparable observations were reported by Jovanović et al. [47]. The visible glass transition temperature absence may be correlated with the polysaccharide nature of microencapsulates [19]. The endothermic peak maximum of these thermal changes dissipated at similar temperatures. According to the DSC analysis, all observed samples indicated good thermal stability (up to 200 °C) of all microencapsulated samples in the temperature region important for food processing and consumption. With carrier addition during the microencapsulation process, the final transition temperatures of the SMEs increased compared to the values of the pure extract.

### 3.3. Chemical Characteristics of SME and Microencapsulates

#### Content of Soluble Free Phenolic Compounds 

As shown in Table 4, the content of total soluble free phenolic compounds (TPC) ranged from 30,622 to 35,506 mg CE/kg, with the highest content in SME. The lower content of TPC in microencapsulates, of about 10 to 14%, compared to the content in SME, was mostly the consequence of dilution by the carrier’s addition. Incomplete extraction due to the strong chemical interaction of functional ingredients and carriers can be excluded as a reason that was confirmed by the results of FTIR analysis (Figure 2). However, it is necessary to point out that the cavity and hollow molecular shape of the CDs make them ideal candidates to decrease phenolic compound loss. This characteristic can provide controlled release properties, which also means difficult extraction [61]. Of the 10 used standards, six individual phenolic acids were detected in SME and all three microencapsulates, namely, 2,3-dihydroxybenzoic acid, vanillic acid, caffeic acid, syringic acid, p-coumaric acid, and ferulic acid (Table 4). In terms of content, caffeic acid, p-coumaric acid, and 2,3-dihydroxybenzoic acid were the most abundant in all investigated samples. However, the content of detected free phenolic acids was low and ranged from 285.9 to 389.4 µg/g in total in SME + MD + HPBCD microparticles and SME, respectively. In general, phenolic acids are mainly present in the insoluble bound form in cereals grains, and the results showed their low bio-accessibility (<1 to 2%) [62,63]. The results of Žilić et al. [62] showed that approximately 76 to 91% of the total ferulic acid and p-coumaric acid in maize grains are in the bound form. Given that the preparation of liquid extract used in these studies was not preceded by the hydrolysis of bioactive compounds from maize by-products, the low content of total phenolic acids in SME and microencapsulates is understandable. The role of free phenolic acids in the prevention of several chronic diseases, as well as aging possibly due to their high antioxidant capacity, is supported by epidemiological evidence [64].

It can be concluded that the largest part of the soluble free phenolic compounds in SME and microencapsulates contains flavonoids, primarily anthocyanins (Table 5). Although the results of FTIR analysis indicated that microencapsulation occurred due to physical incorporation, chemical analyses showed a significantly lower content of total anthocyanins in microencapsulates with HPBCD as a carrier, about 10,222 mg CGE/kg on average. These results are in agreement with those obtained by Tarone et al. [61]. Li et al. [65] examined the light stability of β-CD microcapsules containing mulberry anthocyanins and displayed 1.36 times higher anthocyanin retention for microcapsules compared to nonencapsulated anthocyanins. Due to the use of acidified methanol, 20% and 16% more total anthocyanins were extracted from SME and microencapsulates, respectively, with MD as a carrier. According to the results of Kalušević et al. [66], MD can be more easily dissolved in a broad pH range of solvents compared to other carriers such as gum Arabic and skimmed milk powder, and the highest total phenolic compounds and total anthocyanins were observed in MD-based microparticles. In addition, cyanidin diglucoside from raspberry juice powders was significantly higher in microencapsulates with MD as a carrier (2549.89 µg/g) compared to gum Arabic and waxy starch used as carriers (1935.45 and 1458.81 µg/g, respectively) [67]. It was pointed out that the characteristics of the carriers, their chemical origin, and the bonds within the complex affect the properties of powders important for the stabilization or release of anthocyanins in the food matrix or the digestive tract [68].

One of the great advantages of microencapsulation in the food and pharmacy industries is the high concentration of bioactive compounds that can be carried. For example, the content of total anthocyanins quantified by HPLC in blue popping maize whole-grain flour amounted to 476.26 µg/g [14], while this content per gram of microencapsulates was 8.2- to 9.3-fold higher. This property makes microencapsulates effective vehicles for the targeted administration of specific compounds. On the other hand, microencapsulates do not impair the sensory properties of food products [69]. In our study, identified anthocyanins in the samples were Cy and Pg, conjugated with glucose, and their acylated forms, including mono- and di-malonyl derivatives. A total of eight anthocyanins were detected, i.e., cyanidin-3-glucoside (Cy-3-Glu), pelargonidin-3-glucoside (Pg-3-Glu), two isomers of cyanidin-3-(malonylglucoside) (Cy-3-3Mal-Glu and Cy-3-6Mal-Glu), and three isomers of cyanidin-3-(dimalonyl-β-glucoside) (Cy-3-diMal-Glu). Peonidin-3-glucoside (Pn-3-Glu) was present in traces (Table 5). As shown in Table 5, predominant anthocyanins were acylated forms of cyanidin derivatives whose content accounted for 4223.2, 3516.5, 3450.8, and 3098.3 μg/g, i.e., about 80% of the total content in SME and microencapsulates with MD, HPBCD, and MD + HPBCD as carriers, respectively. The most abundant in all samples was Cy-3-6Mal-Glu, followed by isomers of Cy-3-diMal-Glu (t_R_-16.77) and Cy-3-Glu. The content of Cy-3-6Mal-Glu ranged from 2720.2 to 1936.8 μg/g in SME and microencapsulates with MD + HPBCD, respectively, while the contents of Cy-3-diMal-Glu (t_R_-16.77) and Cy-3-Glu were about 2.7- to 3.5-fold lower. As indicated by the results of numerous authors, the prevalence of nonacylated or acylated forms of anthocyanins in colored maize grains is largely determined by genetic factors [14,70,71]. 

The antioxidant capacity of SME and microencapsulates is shown in Figure 4. The direct or QUENCHER method with ABTS reagents was applied. According to our study, the highest antioxidant capacity (69.14 mmol Trolox Eq/kg) was obtained for the spray-dried maize extract and the mixture of both carriers i.e., 15% MD and 15% HPBCD (58.37 mmol Trolox Eq/kg). The antioxidant capacity of microencapsulates produced only with MD or HPBCD did not differ statistically (63.53 and 62.47 mmol Trolox Eq/kg, respectively). This is in line with the results of Sharayei et al. [72], who used 10% MD and 10% HPBCD as carriers of pomegranate peel extract. Given that the direct procedure that skips all time-consuming solvent extraction and hydrolysis steps was applied, it can be concluded that the antioxidant capacity of microencapsulation depends on each individual compound in the mixture.

In this connection, it should be emphasized that the carrier materials showed various antioxidant capacities [73,74]. However, Kalušević et al. [66] showed that MD, used in this study, did not affect the antioxidant potential of soybean coat extract, while the results of Gil et al. [74] showed that MD had the lowest antioxidant capacity value among other carriers used for the spray-drying process. A higher retention of antioxidant compounds in samples encapsulated by gum arabic than in MD was also observed by Hussain et al. [75]. Gum arabic is a highly branched heteropolymer of sugars containing a small amount of protein covalently linked to the carbohydrate chain. As such, it acts as an excellent film-forming and emulsifying agent capable of entrapping and stabilizing bioactive compounds [75]. In contrast, Kalušević et al. [66] indicated that gum Arabic contributed to the antioxidant capacity of soybean coat extract, with about 60 mmol Trolox Eq/kg. Significant correlations between the content of total phenolic compounds, as well as total anthocyanins and ABTS radical scavenging activities (r^2^ = 0.96 and 0.93, respectively), indicated their contribution to the antioxidant capacity of SME and microencapsulates, as was expected. However, the antioxidant capacity had a highly negative correlation with the in vitro digestibility of SME and microencapsulates (r^2^ = −0.97). The in vitro digestibility values of the tested samples are shown in Figure 4. Microencapsulates with both carriers (15% MD and 15% HPBCD) and the lowest antioxidant capacity had the highest digestibility (73.63%). Conversely, SME had the lowest digestibility. According to our results, it can be assumed that the systems created by encapsulation where the active ingredient is coated by the wall material (MD and HPBCD) protects phenolic compounds/anthocyanins against ABTS radicals [76]. However, during the digestive process, the phenolic compounds/anthocyanins were released due to the swelling and decomposition of microparticles. At the same time, MD and CD derivatives were hydrolyzed by the action of digestive enzymes, and the degree of digestibility of the microencapsulates increased. Maltodextrins have a degree of branching ranging from 5 to 13%, which has an effect on the degree of their digestibility in the small intestine. According to the results of Zhang et al. [77], branched maltodextrins, with a degree of branching of 5–6%, released the highest amount of glucose in 360 min (74–89%), while more highly branched maltodextrins, with shorter average internal chain lengths, released substantially less glucose (60–70%). Regarding β-cyclodextrin, Flourié et al. [78] concluded that it is poorly hydrolyzed in the human small intestine, but that it is fermented in the colon.

## 4. Conclusions

Colored grains are rich sources of anthocyanins that have many beneficial effects on human health and the prevention of various diseases associated with oxidative stress. However, the incorporation of anthocyanins into food is a technological challenge due to their low stability. The extraction process of blue maize waste product was performed by the percolation method, using ethanol, (50% *w*/*w*), with a solid–solvent ratio of 1:10. The stability of extracted valuable bioactive compounds was preserved by the spray-drying technique. With carrier addition during the microencapsulation process, the SMEs’ final transition temperatures increased compared to the values of pure extract. Spray-dried SME + HPBCD had the highest thermal stability, up to 220 °C, indicating good preservation of the polyphenolic structure in the temperature region important for food processing and consumption. Additionally, HPBCD provided significant improvement in the physicochemical characteristics of the powders, rehydration time, moisture content, and particle size distribution. The FTIR method indicated that SME and wall materials were preserved by physical incorporation. Spray-dried maize extract, SME, had the highest antioxidant activity, with the lowest digestibility, which was improved by MD and HPBCD addition. The used biopolymers (MD and HPBCD) had a good impact on the stability of microencapsulates against adverse environmental conditions, which entrapped them inside a coating material. Further study will be oriented to examine the accelerated storage conditions of these microencapsulates for the application of spray-dried blue maize extracts as foods or pharmaceuticals, as well as additional characterization methods that could provide more details of the incorporation of polyphenols into carriers and their bioavailability during digestion.

## Figures and Tables

**Figure 1 foods-12-01870-f001:**
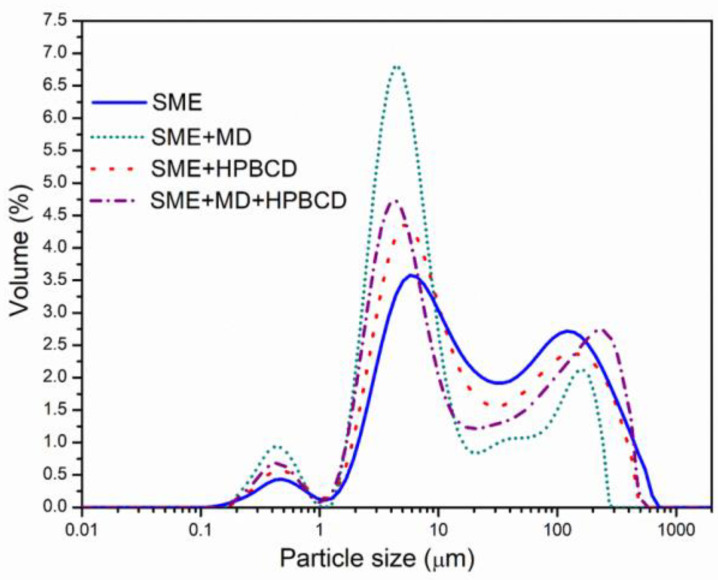
Particle size distribution of SME and microencapsulates.

**Figure 2 foods-12-01870-f002:**
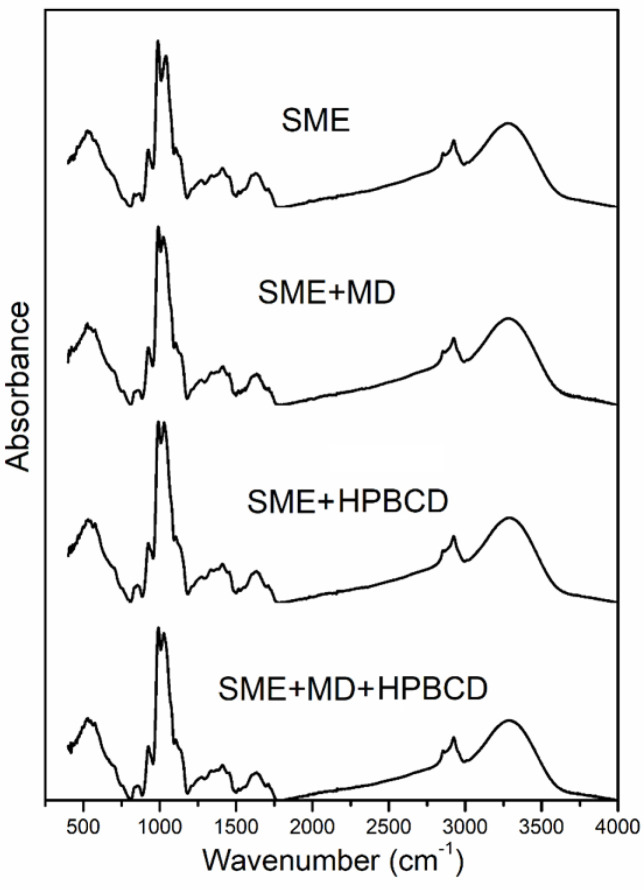
FTIR analysis of SME and microencapsulates.

**Figure 3 foods-12-01870-f003:**
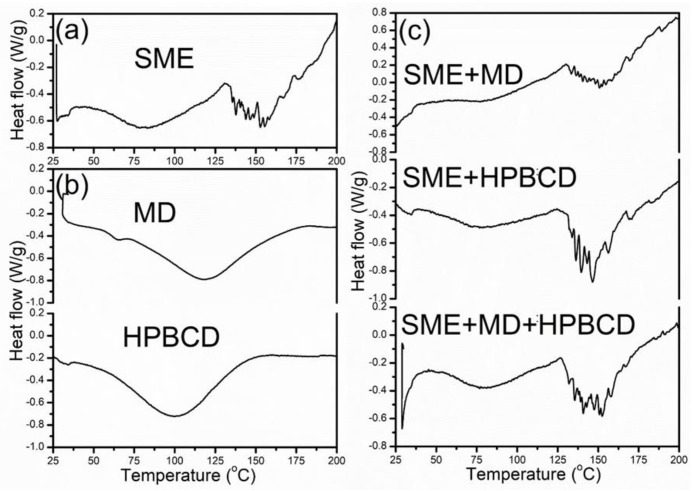
Thermal diagrams of SME, biopolymers, and microencapsulates. (**a**) SME curve; (**b**) MD and HPBCD curves (**c**) SME + MD, SME + HPBCD, and SME + MD + HPBCD curves.

**Figure 4 foods-12-01870-f004:**
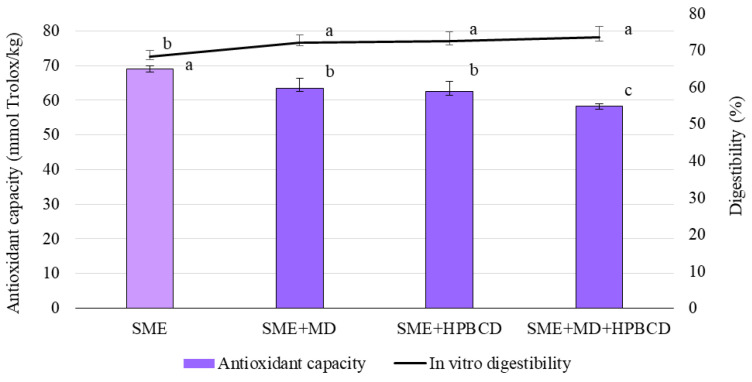
Total antioxidant capacity and in vitro digestibility of SME and its microencapsulates. SME—spray-dried maize extract, MD—maltodextrin, HPBCD—hydroxypropyl-β-cyclodextrin. The vertical bars represent the standard deviation of each data point. Bars with the same letter are not significantly different according to Tukey’s test (α = 0.05%).

**Table 1 foods-12-01870-t001:** Yield, moisture content, bulk and tapped densities, Carr index, Hausner ratio, rehydration, and pH values of SME and microencapsulates.

Samples	Yield	Moisture	Bulk Density	Tapped Density	CI	HR	Rehydration	pH
	(%)	(%)	(g/mL)	(g/mL)			(s)	
SME	73.0 ± 6.1 ^a^	5.40 ± 0.2 ^a^	0.280 ± 0.01 ^c^	0.400 ± 0.02 ^a^	29.29 ± 1.0 ^a^	1.41 ± 0.02 ^a^	224.20 ± 30.7 ^a^	6.62 ± 0.1 ^a^
SME + MD	83.2 ± 6.8 ^a^	4.74 ± 0.4 ^ab^	0.334 ± 0.09 ^bc^	0.377 ± 0.02 ^a^	11.27 ± 3.1 ^b^	1.13 ± 0.04 ^ab^	217.29 ± 22.7 ^a^	6.47 ± 0.1 ^a^
SME + HPBCD	74.53 ± 4.2 ^a^	4.22 ± 0.2 ^b^	0.374 ± 0.02 ^ab^	0.454 ± 0.04 ^a^	17.51 ± 2.5 ^b^	1.21 ± 0.04 ^ab^	153.80 ± 14.5 ^a^	6.48 ± 0.1 ^a^
SME + MD + HPBCD	66.7 ± 1.6 ^a^	4.39 ± 0.1 ^b^	0.260 ± 0.01 ^cd^	0.433 ± 0.03 ^a^	9.17 ± 0.1 ^b^	1.01 ± 0.13 ^b^	133.25 ± 20.2 ^a^	6.55 ± 0.1 ^a^

CI—Carr index, HR—Hausner ratio. SME—spray-dried maize extract, MD—maltodextrin, HPBCD—hydroxypropyl-β-cyclodextrin. Means followed by the same letter within the same column are not significantly different according to Tukey’s test (α = 0.05%).

**Table 2 foods-12-01870-t002:** The particle size of SME and microencapsulates.

Samples	d_10_ *	d_50_ **	d_90_	Span ***	D [4.3]	D [3.2]	Uniformity
SME	3.33± 0.31 ^a^	21.33 ± 3.43 ^a^	237.15 ± 17.9 ^a^	10.96 ± 1.05 ^c^	78.53 ± 8.59 ^a^	5.59 ± 0.36 ^a^	3.32 ± 0.17 ^b^
SME + MD	2.22 ± 0.39 ^b^	4.72 ± 0.47 ^c^	128.62 ± 8.2 ^b^	20.67 ± 2.69 ^ab^	32.34 ± 2.32 ^b^	3.11 ± 0.38 ^c^	4.72 ± 0.59 ^b^
SME + HPBCD	2.81 ± 0.49 ^ab^	12.28 ± 2.12 ^b^	206.50 ± 18.7 ^a^	16.01 ± 2.64 ^bc^	63.00 ± 11.28 ^a^	4.69 ± 0.68 ^ab^	4.52 ± 0.28 ^b^
SME + MD + HPBCD	2.45 ± 0.42 ^ab^	10.73 ± 1.21 ^bc^	257.14 ± 37.3 ^a^	23.73 ± 3.71 ^a^	75.02 ± 7.14 ^a^	4.06 ± 0.47 ^bc^	6.57 ± 1.02 ^a^

* d_10_, d_50_, d_90_ signify the sizes where 10%, 50%, and 90% of the microparticles are smaller than the remaining particles, ** mean diameter, *** calculated as (d_90_ − d_10_)/d_50_. SME—spray-dried maize extract. MD—maltodextrin, HPBCD—hydroxypropyl-β-cyclodextrin. Means followed by the same letter within the same column are not significantly different according to Tukey’s test (α = 0.05%).

**Table 3 foods-12-01870-t003:** The thermal characteristics of spray-dried SME and microencapsulates.

Samples	T1	T2	T3	T4	∆H1	∆H2	∆H3	∆H4
SME	86.18 ± 14.1 ^a^	150.55± 11.9 ^a^	197.06 ± 23.7 ^a^	261.11 ± 42.6 ^a^	70.68	145.91	6.51	60.73
SME + MD	86.26 ± 4.45 ^a^	154.18± 16.1 ^a^	241.00 ± 16.8 ^a^	275.00 ± 22.8 ^a^	41.85	140.29	5.66	12.67
SME + HPBCD	76.87 ± 11.9 ^a^	146.57± 14.7 ^a^	236.28 ± 39.4 ^a^	276.00 ± 43.9 ^a^	35.89	161.08	9.70	16.78
SME + MD + HPBCD	83.53 ± 4.9 ^a^	152.48± 20.2 ^a^	234.00 ± 40.9 ^a^	234.00 ± 31.2 ^a^	40.63	70.62	4.96	5.14

T—temperature transition, ∆H—enthalpy change. SME—spray-dried maize extract, MD—maltodextrin, HPBCD—hydroxypropyl-β-cyclodextrin. Means followed by the same letter within the same column are not significantly different according to Tukey’s test (α = 0.05%).

**Table 4 foods-12-01870-t004:** Contents of total free phenolic compounds (mg CE/kg) and detected free phenolic acids (µg/g) in SME and microencapsulates.

Samples	TPC	DHBA	VA	CAFA	SYRA	*p*-CA	FA	TPA
SME	35506 ± 1800 ^a^	96.47 ± 0.7 ^a^	14.89 ± 0.1 ^a^	117.64 ± 1.6 ^a^	25.91 ± 0.6 ^a^	105.32 ± 0.5 ^a^	29.18 ± 0.8 ^a^	389.4 ^a^
SME + MD	32211 ± 1085 ^bc^	78.04 ± 1.7 ^b^	12.38 ± 1.1 ^ab^	81.27 ± 1.1 ^b^	25.40 ± 0.6 ^a^	85.70 ± 0.9 ^b^	28.29 ± 0.8 ^a^	311.1 ^b^
SME + HPBCD	31308 ± 708 ^c^	81.29 ± 0.9 ^b^	10.53 ± 0.7 ^b^	79.02 ± 1.1 ^b^	22.98 ± 0.5 ^b^	86.22 ± 0.7 ^b^	27.70 ± 0.6 ^a^	307.6 ^b^
SME + MD + HPBCD	30622 ± 361 ^c^	82.29 ± 1.4 ^b^	10.35 ± 0.5 ^b^	79.2 ± 0.8 ^b^	18.93 ± 0.8 ^c^	77.96 ± 1.2 ^c^	17.23 ± 1.6 ^b^	285.9 ^b^

TPC—total phenolic compounds, DHBA—2,3-dihydroxybenzoic acid, VA—vanillic acid, CAFA—caffeic acid, SYRA—syringic acid, p-CA—p-coumaric acid, FA—ferulic acid, TPA—total detected phenolic acid. SME—spray-dried maize extract, MD—maltodextrin, HPBCD—hydroxypropyl-β-cyclodextrin, n.d.—not detected, Means followed by the same letter within the same column are not significantly different according to Tukey’s test (α = 0.05%).

**Table 5 foods-12-01870-t005:** Content of total anthocyanins (mg CGE/kg) and detected individual anthocyanins (µg/g) in SME and microencapsulates.

Compounds *	t_R_	SME	SME + MD	SME + HPBCD	SME + MD + HPBCD
TANs		12,846 ± 84 ^a^	12,182 ± 77 ^ab^	10,802 ± 152 ^bcd^	9642 ± 451 ^d^
Cy-3-Glu	8.87	770.4 ± 6.1 ^a^	657.4 ± 29.7 ^ab^	742.6 ± 38.2 ^ab^	626.9 ± 38.2 ^b^
Pg-3-Glu	10.73	244.3 ± 1.1 ^a^	225.4 ± 4.9 ^b^	234.9 ± 5.1 ^ab^	184.9 ± 2.8 ^c^
Pn-3-Glu	11.70	trace	trace	trace	trace
Cy-3-3Mal-Glu	12.08	326.3 ± 6.9 ^a^	239.2 ± 4.1 ^bc^	238.4 ± 3.7 ^bc^	187.3 ± 22.7 ^c^
Cy-3-6Mal-Glu	13.61	2720.2 ± 25.9 ^a^	2258.1 ± 13.9 ^bc^	2188.1 ± 3.1 ^c^	1936.8 ± 33.4 ^d^
Cy-3-diMal-Glu	15.63	309.7 ± 0.4 ^a^	246.6 ± 1.1 ^bc^	242.9 ± 5.3 ^bc^	205.4 ± 9.4 ^c^
Cy-3-diMal-Glu	16.19	80.5 ± 8.1 ^a^	56.2 ± 7.1 ^b^	47.5 ± 5.4 ^bc^	34.4 ± 2.5 ^c^
Cy-3-diMal-Glu	16.77	786.5 ± 33.9 ^a^	716.5 ± 18.6 ^a^	733.9 ± 22.7 ^a^	734.5 ± 29.7 ^a^
Total detected ANs		5237.8 ^a^	4399.4 ^b^	4428.3 ^b^	3910.2 ^c^
*Total cyanidin derivatives*					
Non-acylated		770.4 ^a^	657.6 ^ab^	742.7 ^ab^	627.0 ^b^
Acylated		4223.2 ^a^	3516.5 ^bc^	3450.8 ^c^	3098.3 ^d^

TANs—total anthocyanins, Ans—anthocyanins, n.d.—not detected, Cy-3-Glu—cyanidin-3-glucoside, Pg-3-Glu—pelargonidin-3-glucoside, Pn-3-Glu-peonidin-3-glucoside, Cy-3-3Mal-Glu and Cy-3-6Mal-Glu—isomers of cyanidin-3-(malonylglucoside), Cy-3-diMal-Glu—isomers of cyanidin-3-(dimalonyl-β-glucoside). SME—spray-dried maize extract, MD—maltodextrin, HPBCD—hydroxypropyl-β-cyclodextrin, Means followed by the same letter within the same row are not significantly different according to Tukey’s test (α = 0.05%).

## Data Availability

All data generated or analysed during this study are included in this published article.

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
