# Peer review of "Microencapsulates of Blue Maize Polyphenolics as a Promising Ingredient in the Food and Pharmaceutical Industry: Characterization, Antioxidant Properties, and In Vitro-Simulated Digestion"

_foods, 2023, doi:10.3390/foods12091870_

Round 1

Reviewer 1 Report

The manuscript entitled " Microencapsulates of phenolic compounds from blue maize processing by-product with potential application in food and pharmaceutical industry" report quite interesting results obtained by systematic investigation. The content of the paper is quite interesting and promises some commercial application. I have the following suggestions for improvement of the paper.

The idea of the article is good. It is almost well designed. But its analysis needs to be completed. However, a few points to improve the current format of the article will be mentioned below:

The title is general, please make it more specialized and in more detail

Abstract: The abstract should be more informative by giving real results rather than elastic sentences. Important and main contents should be given. Support the results with some quantitative data. In the abstract or conclusion mention which condition showed comparatively improved characteristics.

Please choose keywords in such a way that they are not mentioned in the title in addition to helping to understand the concept of the research.

Introduction: what is the novelty of your work?

The introduction of the manuscript is heedlessly written. It should be written completely. The introduction needs to be revised and improved with various previous such as https://doi.org/10.1186/s40538-021-00220-z

There are several spelling and grammar mistakes in the manuscript. Many loose sentences without providing actual meaning have been found. Read thoroughly and correct them.

The purpose of the research should be well explained at the end of the introduction.

Methods: the methods describe well.

Discussion section: This part needs more specific detailed comparative studies. Please compare with similar works after presenting each result.

Conclusion: what is the future of your findings? Conclusion is not insightful, what are suggestions?

There are several spelling and grammar mistakes in the manuscript. Many loose sentences without providing actual meaning have been found. Read thoroughly and correct them.

Author Response

Dear Reviewer, please see attached document.

Reviewer 2 Report

The article is about the microencapsulates of phenolic compounds from blue maize processing by-products with potential application in the food and pharmaceutical industry. Anthocyanins were extracted from the waste product of blue maize processing. The extracted compounds were spray-dried using maltodextrin and hydroxypropyl-β-cyclodextrin as wall material. It can be considered as applied research, and I think the subject is overall interesting. However, the manuscript needs minor changes for improvement.  

Line 78, 79, 80, 81: Split your long sentences into shorter ones; The aim of the present study was to develop microencapsulation systems for phenolic compounds from blue maize processing by-product using conventional wall material, MD, in combination with novel one, HPBCD, as well as to characterize microencapsulates in order to obtain powders with appropriate functional, organoleptic and biological characteristics, uniform in size, which could be applied in food and pharmaceutical industry.

Introduction: State the reason for using maltodextrin and hydroxypropyl-β-cyclodextrin as wall material. Their compatibility issues with respect to the phenolic compounds

Line 176: Pl. mention the detailed conditions for spectral analysis of beads including the pretreatment of beads (with Kbr or other material), resolution, scan speed, total number of scans, pretreatment of data

Line 176-199: References are missing. Are these claims based on your own investigations? Where are the literature data?

Line 204-205: Correct the formula of Na2CO3: After 5 min, the mixture was neutralized with 1.25 mL of 20% aqueous Na2CO3 solution

Line 284: Remove extra space before results and discussion.

Line 299: Discuss the results of moisture content in detail

References: Update the references

The article is about the microencapsulates of phenolic compounds from blue maize processing by-products with potential application in the food and pharmaceutical industry. Anthocyanins were extracted from the waste product of blue maize processing. The extracted compounds were spray-dried using maltodextrin and hydroxypropyl-β-cyclodextrin as wall material. It can be considered as applied research, and I think the subject is overall interesting. However, the manuscript needs minor changes for improvement.  

Line 78, 79, 80, 81: Split your long sentences into shorter ones; The aim of the present study was to develop microencapsulation systems for phenolic compounds from blue maize processing by-product using conventional wall material, MD, in combination with novel one, HPBCD, as well as to characterize microencapsulates in order to obtain powders with appropriate functional, organoleptic and biological characteristics, uniform in size, which could be applied in food and pharmaceutical industry.

Introduction: State the reason for using maltodextrin and hydroxypropyl-β-cyclodextrin as wall material. Their compatibility issues with respect to the phenolic compounds

Line 176: Pl. mention the detailed conditions for spectral analysis of beads including the pretreatment of beads (with Kbr or other material), resolution, scan speed, total number of scans, pretreatment of data

Line 176-199: References are missing. Are these claims based on your own investigations? Where are the literature data?

Line 204-205: Correct the formula of Na2CO3: After 5 min, the mixture was neutralized with 1.25 mL of 20% aqueous Na2CO3 solution

Line 284: Remove extra space before results and discussion.

Line 299: Discuss the results of moisture content in detail

References: Update the references

Author Response

Dear reviewer, please see attached document.

Reviewer 3 Report

This manuscript investigates microencapsulates of phenolic compounds from blue maize processing by-product, and its potential application in food and pharmaceutical industry. The manuscript is interesting and well organized, but still needs some revision:

1.The authors should check the format of whole manuscript, such as Line 178 “ 4000 cm-1”, Table 2 “D90”, Lines 258”In vitro ”, etc.

2.The related reference should be cited in method 2.6.1.

3. The enzyme activity in pepsin and pancreatin as well as in the final solution should be added. (Line 268-270).

4. “2.1. Ingredients” is better to revised as “2.1. materials”, and the involved materials e.g., pepsin and pancreatin should added.

5. The authors should discuss the comparison of different results from those published papers. Such as references [43], why the reported the opposite result? Line 306-308

6. “...in this case ABTS radicals...” need to be checked.

7. The data in table 5 should be revised as “ average ± SD”.

8. In conclusion, please discuss the lack of this study and the design of future study.

9. Some of the cited references were too old, and some of the cited reference is needed to be delected.

Author Response

(The authors gave the same response as above.)

Reviewer 4 Report

I think that this research based on the protection of anthocyanins through the addition of two polysaccharides used as carriers is not very original. In general, the work is well developed and well written. I believe that the novelty of the work is not properly sustained in the manuscript. The strategy used for the protection of these active compounds is very studied in literature. I believe that the results obtained are consistent with the strategy used. The properties of the powders obtained in terms of moisture, density properties, rehydration, particle size, etc. are consistent and expected according to the carrier used. I also believe that the amount of carrier used would not make it possible to obtain microencapsulates, or at least this should be studied further in order to confirm it. I strongly believe that in this work it is necessary to add the study of the stability of anthocyanins under different environmental conditions for different periods of time. In this way, the success or not of the initial objective, which was the microencapsulation of anthocyanins to protect them from their low stability, would be evidenced. In addition, some suggestions and comments are enclosed in the pdf file.

Author Response

(The authors gave the same response as above.)

Round 2

Reviewer 4 Report

I believe that the proposed changes have been made successfully. However, in my previous review I suggest to carry out extra experiments that study the following aspects:

- The formation of anthocyanin inclusion complexes in the beta-CD cavities

- Stability studies of encapsulated anthocyanins under different environmental conditions and for different periods of time

- bioavailability studies after the digestion process.

The authors respond that these studies are planned for future publications. I still think that these studies are essential and would complete this work, so I leave the final decision to the editor. If the editor considers that these studies are not essential for this publication, this manuscript can be accepted.